# Plant-Based Diets for Cardiovascular Safety and Performance in Endurance Sports

**DOI:** 10.3390/nu11010130

**Published:** 2019-01-10

**Authors:** Neal D. Barnard, David M. Goldman, James F. Loomis, Hana Kahleova, Susan M. Levin, Stephen Neabore, Travis C. Batts

**Affiliations:** 1Adjunct and Clinical Faculty, Department of Internal Medicine, George Washington University School of Medicine and Health Sciences, Washington, DC 20016, USA; nbarnard@pcrm.org (N.D.B.); jloomis@pcrm.org (J.F.L.); sneabore@pcrm.org (S.N.); 2Department of Clinical Research, Physicians Committee for Responsible Medicine, Washington, DC 20016, USA; hkahleova@pcrm.org; 3Barnard Medical Center, Washington, District of Columbia, DC 20016, USA; 4Private Practice, Bend, OR 97702, USA; davidmichaelgoldman@gmail.com; 5Wilford Hall Ambulatory Surgical Center, San Antonio, TX 78236, USA; doctorbatts11@gmail.com

**Keywords:** nutrition, exercise, athletic performance, vegan, vegetarian

## Abstract

Studies suggest that endurance athletes are at higher-than-average risk for atherosclerosis and myocardial damage. The ability of plant-based regimens to reduce risk and affect performance was reviewed. The effect of plant-based diets on cardiovascular risk factors, particularly plasma lipid concentrations, body weight, and blood pressure, and, as part of a healthful lifestyle, reversing existing atherosclerotic lesions, may provide a substantial measure of cardiovascular protection. In addition, plant-based diets may offer performance advantages. They have consistently been shown to reduce body fat, leading to a leaner body composition. Because plants are typically high in carbohydrate, they foster effective glycogen storage. By reducing blood viscosity and improving arterial flexibility and endothelial function, they may be expected to improve vascular flow and tissue oxygenation. Because many vegetables, fruits, and other plant-based foods are rich in antioxidants, they help reduce oxidative stress. Diets emphasizing plant foods have also been shown to reduce indicators of inflammation. These features of plant-based diets may present safety and performance advantages for endurance athletes. The purpose of this review was to explore the role of nutrition in providing cardioprotection, with a focus on plant-based diets previously shown to provide cardiac benefits.

## 1. Introduction:

The Academy of Nutrition and Dietetics, Dietitians of Canada, and the American College of Sports Medicine have determined that “the performance of, and recovery from, sporting activities are enhanced by well-chosen nutrition strategies” [1]. In this review, we examine physiological effects of plant-based diets that present potential safety and performance advantages in endurance sports. These include improvements in cardiovascular risk factors, improved blood flow, leaner body composition, reduced oxidative stress, reduced inflammation, and improved glycogen storage, among others. Several studies have shown that plant-based dietary patterns have particular benefits for heart health. A low-fat, vegetarian diet, along with other healthful lifestyle changes, has been shown to reverse arterial plaque [2,3,4]. Compared with meat eaters, vegetarians are 32% less likely to develop coronary heart disease [5]. In this review, the terms “plant-based” and “vegan” will be used interchangeably to refer to a diet without animal-derived products. Variants of vegan diets, e.g., vegetarian diets that may include dairy products or eggs, will be identified when relevant.

Arterial changes that contribute to atherosclerosis can begin early in life, even in utero [6]. By age 10 to 14 years, the majority of American children have fatty streaks in the left anterior descending coronary artery, and more than five percent have more advanced coronary disease [7]. Autopsies of U.S. soldiers who died in the Korean War revealed severe coronary atherosclerosis (75% to 90% narrowing) in 6.4% at a mean age of 20.5 years [8]. A similar study of autopsies of soldiers who died in Operations Enduring Freedom and Iraqi Freedom/New Dawn between October 2001 and August 2011 showed that, at a mean age of 25.9 years, 8.5% had coronary atherosclerosis [9]. By age 20, approximately 10% of the population in developed countries have advanced atherosclerotic lesions in the abdominal aorta, reducing blood flow and contributing to disc degeneration and lower back pain [10].

Athletes are not immune to atherosclerosis or to cardiac events [11]. Surprisingly, endurance athletes may have more advanced atherosclerosis and more myocardial damage, compared with sedentary individuals, particularly as they age. In a 2017 study in the United Kingdom, coronary plaques were found in 44% of middle-aged and older endurance athletes engaged in cycling or running, compared with 22% of sedentary controls (*p* = 0.009) [12]. Similarly, a study of 50 men who had run at least 25 consecutive Twin Cities Marathons (Minneapolis, MN, USA) found the runners to have increased total plaque volume, calcified plaque volume, and non-calcified plaque volume, compared with 23 sedentary controls [13]. In a study of active German marathon runners ≥ 50 years of age, myocardial damage, as measured by magnetic resonance imaging, was found in 12% of active runners, compared with 4% of sedentary controls [14]. The degree of myocardial damage was predicted by the number of marathons run [15].

Sports-related sudden cardiac deaths are uncommon, but increase with age. In older athletes, these deaths are typically attributable to coronary artery disease (more than 80% of cases), with additional cases attributable to hypertrophic cardiomyopathy, arrhythmogenic right ventricular dysplasia, myocarditis, and valvular heart disease [16].

These studies show that well-trained athletes are at significant risk for atherosclerosis and myocardial damage. What they do not show is whether these changes are the consequences of athletic activity or of the foods often used to fuel it. To the extent that increased consumption of animal products supplies the energy for increased athletic activity, their saturated fat and cholesterol and relative absence of antioxidants and fiber may contribute to atherosclerotic changes.

Apart from increasing the risk of cardiac events, atherosclerosis may also narrow arteries to the legs, the brain, and other parts of the body, reducing blood flow and potentially impairing performance. While this is particularly evident in diagnosed peripheral artery disease [17], it may also be a factor, at least theoretically, for athletes with subclinical atherosclerotic disease.

Plant-based diets address key contributors to atherosclerosis: dyslipidemia, elevated blood pressure, elevated body weight, and diabetes, each of which is briefly discussed in the following paragraphs.

Regarding plasma lipids, dyslipidemia is a major contributor to arterial disease and is promoted by diets rich in saturated fat and, to a lesser degree, dietary cholesterol, as noted above. Dairy products and meat are the leading sources of saturated fat, and exclusion of these products predictably improves plasma lipid profiles [18], an effect that can be accentuated by the specific inclusion of soluble fiber (e.g., oats, barley, or beans), soy protein, almonds, and sterol-containing margarines. Combining these elements in a “portfolio” diet, University of Toronto researchers lowered low-density lipoprotein cholesterol levels by nearly 30 percent in four weeks [19]. Animal products are not the only offenders. Trans fats also have harmful effects on plasma lipids and pose cardiovascular risks [20].

Regarding blood pressure, vegan and vegetarian diets reduce both systolic and diastolic blood pressure, which appears to be a consequence of reduced blood viscosity, increased blood potassium, and weight loss [21]. Lower blood pressure reduces the risk of atherosclerotic changes.

Obesity is a risk factor for cardiovascular disease. Vegetarian, especially vegan, diets reduce body fat, even in the absence of intentional limitations on calories or portion sizes [22].

Regarding glycemic control, plant-based diets boost insulin sensitivity [23]. This is important for reducing the risk of type 2 diabetes and improving glycemic control in individuals with diabetes, which is a major contributor to atherosclerosis.

Each of these factors is improved by plant-based diets. As part of an overall healthful lifestyle, a low-fat vegetarian diet has been shown to reverse coronary atherosclerosis, increasing blood flow and reducing the risk of coronary events [2]. Specifically, the Lifestyle Heart Trial limited the use of animal products to egg whites and 1 cup of nonfat milk or yogurt daily. Because flow through a vessel is proportional to the fourth power of its radius, even small changes in the arterial diameter can lead to major changes in blood flow.

## 2. The Role of Diet in Athletic Performance

Apart from their role in cardiovascular health, plant-based diets have other physiological effects that may offer performance advantages. These include a leaner body mass, ease of glycogen storage, improved tissue oxygenation, reduced oxidative stress, and reduced inflammation.

### 2.1. Leaner Body Mass

As noted above, plant-based diets reduce body fat, an effect that not only reduces atherosclerotic risk but may also be directly beneficial for athletic performance. The reduction in body fat is mainly due to the low fat content and high fiber content of these diets, traits that reduce the energy density of meals, with a corresponding reduction in energy intake. However, even without changes in body weight, transitioning to a meat-free diet can significantly reduce body fat as measured by skinfold thickness and waist:height ratio [24]. Plant-based diets also influence postprandial energy expenditure. In a 2005 study, the use of a low-fat vegan diet for 14 weeks increased postprandial energy expenditure by 16% [23]. This effect may be due to changes in mitochondrial activity. The number and activity of mitochondria in muscle cells and other body tissues are not constant; rather, they change depending on the diet. In a study in which volunteers were fed a 50%-fat diet, mitochondrial biogenesis was significantly reduced within 3 days [25].

High-fat diets may also act on cellular metabolism indirectly through their effects on the gut microbiome. Gut bacteria produce endotoxins that can enter the bloodstream and, in turn, influence cellular metabolism. High-fat diets appear to disrupt the intestinal barrier to the passage of endotoxins. In a 5-day experiment in human volunteers, a 55%-fat diet led to a marked increase in circulating endotoxins and, in turn, to a significant impairment of postprandial cellular glucose oxidation [26].

These findings suggest that high-fat diets quickly disrupt cellular metabolism, reducing energy expenditure, while a low-fat, plant-based diet has the opposite effect, increasing postprandial energy expenditure.

As we have seen, plant-based diets reduce body fat by reducing dietary energy density and increasing postprandial metabolism. However, there is also a greater metabolic cost of converting dietary carbohydrate to body fat, compared with converting dietary fat to body fat. As a result, low-fat diets are more effective than calorie-matched low-carbohydrate diets for reducing body fat. In a crossover trial including 19 overweight adults given two isocaloric diets for six days each, a low-fat diet resulted in a significantly greater loss of body fat (−89 g of body fat per day) compared with a low-carbohydrate diet (−53 g of body fat per day, *p* = 0.002) [27].

Eliminating excess body fat not only reduces atherosclerotic and metabolic risks, it also boosts endurance. Specifically, reduced body fat is associated with increased submaximal and maximal aerobic capacity [28,29]. An athlete with a higher VO2 max relative to their body weight will have better endurance and will outperform an athlete with a lower value [30,31], and the effect of diet on VO2 max relative to body weight is important, not only for high-level competitors, but for individuals who are not trained athletes. In a study of 31 overweight women, the loss of 9.2 kg of body fat was accompanied by a 15% increase in VO2 max relative to body weight [28]. Even in the absence of weight loss, a vegetarian, mostly vegan, dietary pattern has been shown to reduce visceral fat and increase VO2 max [32].

### 2.2. Facilitating Glycogen Storage

Carbohydrate is the primary energy source during moderate and high-intensity aerobic exercise, and endurance is enhanced by a high-carbohydrate intake, not only immediately before athletic events, but over the long term [33]. Many athletes, however, have eating patterns that are deficient in carbohydrate, putting them at risk for an overly rapid depletion of glycogen from the muscle and liver and early fatigue. A 2016 study of athletes participating in full and half Ironman triathlons, winter triathlons, and winter pentathlons showed that fewer than half (46%) reported meeting the recommended carbohydrate intake for athletes training 1–3 h per day (≥6 g/kg body weight per day) [34]. Because grains, legumes, and root vegetables are rich in complex carbohydrate, individuals who begin plant-based diets typically increase their intake of healthful carbohydrate [35].

### 2.3. Reduced Blood Viscosity and Increased Tissue Oxygenation

A key factor in oxygen delivery to the muscles and other tissues is blood viscosity (resistance to flow, or “thickness”), which is a function of plasma viscosity and packed cell volume [36]. Generally speaking, lowering blood viscosity will improve blood flow and thereby improve athletic performance [37]. In the course of athletic activity, however, the passage of fluid from the bloodstream into the tissues leads to hemoconcentration. Gradually rising blood viscosity causes a progressive loss of tissue oxygenation, degrading performance [38].

Aerobic training increases blood volume, and because it increases plasma volume to a greater extent than red cell mass, aerobic training reduces blood viscosity [38]. However, plasma viscosity is also influenced by food choices. Because plants are typically low in saturated fat and devoid of cholesterol, vegetarian diets reduce plasma lipid concentrations [18], leading to reduced viscosity. In a study comparing 48 individuals following vegetarian eating patterns and 41 matched controls, plasma viscosity, packed cell volume, and blood viscosity were lower in vegetarians, and the stricter the avoidance of animal products, the greater the observed differences [39]. Individuals excluding meats entirely had significantly lower blood viscosity, compared with those having occasional meat (less than once a week). These observations were initially identified as an explanation for the lower blood pressure and lower prevalence of hypertension that are commonly observed among those following vegetarian diets. However, reduced blood viscosity also improves tissue oxygenation, potentially improving athletic performance.

Blood flow also depends on arterial flexibility. Healthy arteries expand with the pressure of a pulse wave (compliance) and then return to their previous diameter when the wave has passed (elasticity). Over time, hypertension, dyslipidemia, and chronically elevated glucose levels associated with diabetes can injure the artery wall, leading to inflammation and matrix remodeling, making arteries “stiffer” [40,41].

Vasoactivity is also influenced by diet habits. University of Maryland investigators assessed brachial artery flow-mediated vasodilation in a crossover study in which 18 participants followed a low-fat vegetarian (Ornish) diet, a low-carbohydrate, high-fat (Atkins) diet, and a modified low-carbohydrate, high-fat (South Beach) diet, for four weeks each, adjusting energy intake to prevent weight loss. Participants were relatively young (mean age 31 years) and slim (mean body mass index 22.6 kg/m^2^.) The vegetarian diet improved brachial artery flow-mediated vasodilation, compared with the low-carbohydrate diet, while the modified low-carbohydrate diet yielded results between the two. The higher the saturated fat intake, the greater the impairment of flow-mediated vasodilation [42].

Arterial compliance can even be impaired by a single high-fat meal. The Maryland researchers measured the effects of a high-fat meal on artery function in 10 healthy volunteers. The test meal consisted of a McDonald’s Egg McMuffin, a Sausage McMuffin, two hash brown patties, and a drink, providing 900 calories, 50 g of fat, 14 g of saturated fat, and 255 mg of cholesterol. Flow-dependent vasoactivity fell from a pre-meal value of 21% to 11% at two hours and was still low (11%) at four hours postprandially [40].

Although meals rich in animal fats impair arterial compliance, some added oils may have similar short-term effects. A single-meal experiment using a carrot cake and a milk shake prepared with coconut oil (which is high in saturated fat) demonstrated impaired arterial compliance, compared with the same meal prepared with safflower oil (which is high in polyunsaturated fat) [43]. Studies using olive oil (rich in monounsaturated fat) have yielded mixed results—some showed an impairment of flow-mediated vasodilation, others did not [44,45,46]. Overall, these studies suggest that, while animal fats are particularly harmful for arterial flexibility, there is a benefit from meals prepared from vegetables, grains, legumes, and fruits, without animal products or added oils.

The factors described above—blood viscosity, arterial diameter, and arterial compliance and elasticity—are all influenced by food choices and may all be expected to affect improve tissue oxygenation, endurance, and performance.

As we have seen, athletic activity depends on good circulation to provide oxygen and nutrients and carry away metabolic waste products. Blood flow to the muscles is influenced by blood viscosity, as well as by arterial caliber, compliance, and elasticity, all of which are influenced by food choices.

### 2.4. Reduced Oxidative Stress

Exercising muscles produce reactive oxygen species (free radicals). These free radicals result from the normal function of mitochondria and other intracellular organelles during exercise, as well as from cellular responses to tissue damage [47,48]. When the production of reactive oxygen species exceeds the body’s ability to neutralize free radicals through endogenous and exogenous antioxidants, the result is called oxidative stress. At low levels, oxidative stress upregulates antioxidant defenses [49] and boosts the immune response [50]. However, free radical production that greatly exceeds the neutralizing ability of antioxidant defenses can result in damage to DNA (leading to mutations), to plasma lipids (leading to atherosclerosis), and to proteins (leading to cell damage and accelerated aging). Exercise-related oxidative stress can also lead to muscle fatigue, reduced athletic performance, and impaired recovery [51].

Compared with omnivores, people following vegan and vegetarian diets have increased antioxidant activity, due to higher intakes of vitamin C, vitamin E, beta-carotene, and other antioxidants [52], as well as to higher antioxidant enzyme production [53]. Researchers have also found potentially beneficial effects of specific antioxidant-rich foods on exercise outcomes, notably beets [54], allium vegetables (e.g., garlic, onions, and leeks) [55], and cherry juice [56]. Antioxidant supplements, such as vitamin E, vitamin C, beta carotene, and glutathione, have been shown to reduce oxidative stress, but some researchers have suggested that they may delay muscle recovery, prevent some of the positive health effects of exercise, and block the improvement of insulin sensitivity associated with exercise [22,57].

### 2.5. Reduced Inflammation

Food choices may help reduce inflammation. Although regular exercise reduces the chronic inflammation associated with obesity, metabolic syndrome, and type 2 diabetes [58,59], acute bouts of intense exercise can elicit an inflammatory response and contribute to delayed-onset muscle soreness. This condition, manifested by pain, reduced muscular performance, and impaired recovery, is more common in untrained individuals and after eccentric muscle activity [60].

A plant-based diet appears to be a helpful part of a strategy to reduce inflammation. In a 2017 meta-analysis of 18 prior studies, vegetarian diets consumed over a two-year period were shown to reduce serum concentrations of C-reactive protein (a marker of inflammation), suggesting an anti-inflammatory effect of plant-based foods [59]. The anti-inflammatory benefits of plant-based diets may stem from (1) their antioxidant content, (2) the absence of products that may be inflammatory or sensitizing, or (3) the absence of pro-inflammatory fats. A few studies have examined the possibility that specific foods with antioxidant activity (e.g., tart cherries [61], pomegranates [62], blueberries [63], blackcurrants [64], and watermelon [65]) may decrease post-exercise inflammation and facilitate recovery.

The pro- or anti-inflammatory effects of foods may also influence joint symptoms. In the Nurses’ Health Study, including 3690 participants, as total red meat consumption increased, C-reactive protein, hemoglobin A1c (an indicator of glycemic control), and stored iron (iron excess is associated with heart disease, cancer, and diabetes) also increased [66]. It should be noted that, in studies of processed meats, it is impossible to separate the effects of meat itself from additives used in its preparation. Osteoarthritis, once attributed to “wear and tear,” is now known to have an important inflammatory component, and is aggravated by overweight and diabetes [67]. Psoriatic arthritis and many other conditions are similarly known to be manifestations of inflammatory processes. In the Adventist Health Study, individuals who ate meat at least once per week had a higher prevalence (49% and 43% higher prevalence for women and men, respectively) of degenerative arthritis and soft tissue disorders compared with individuals who avoided meat [68]. In individuals with rheumatoid arthritis, several studies have demonstrated that vegan and vegetarian diets reduce C-reactive protein, as well as both subjective and objective signs of arthritis [69,70,71,72,73]. A vegan diet has also been shown to have an anti-inflammatory effect (as evidenced by reductions in C-reactive protein) in patients with and without coronary artery disease [74,75,76].

## 3. Nutrient Adequacy

Nutrient adequacy is an important consideration with any dietary regimen. Individuals who change from omnivorous to plant-based diets typically improve their overall nutrition, as judged by the alternate healthy eating Index, developed by Harvard University researchers [77]. The reason for this is that fruits, vegetables, beans, and whole grains tend to be high in vitamins, minerals, and fiber, very low in saturated fat, and devoid of cholesterol.

While protein adequacy is a frequently raised question, surveys show that virtually all endurance athletes meet recommended protein intakes [34], and a varied diet of plant foods easily provides adequate amounts of all essential amino acids for athletes [35].

Calcium is abundant in many plant foods, especially green leafy vegetables and legumes. Those seeking extra calcium will find plant-based dairy alternatives to be a convenient source. Perhaps surprisingly, iron intake is often higher on plant-based diets than on meat-containing diets, due to the large iron content of green vegetables and legumes. Among vegetarians, serum ferritin levels are typically within the normal range. Vitamin B12, which is essential for nerve function and blood cell formation, must be supplemented on a plant-based diet [35].

## 4. Conclusions

Plant-based diets play a key role in cardiovascular health, which is critical for endurance athletes. Specifically, these diets improve plasma lipid concentrations, blood pressure, body weight, and blood glucose control, and, as part of a healthful lifestyle, have been shown to reverse atherosclerosis. The possibility that such diets may also contribute to improved performance and accelerated recovery in endurance sports is raised by their effects on blood flow, body composition, antioxidant capacity, systemic inflammation, and glycogen storage. These attributes provide a scientific foundation for the increased use of plant-based diets by endurance athletes.

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
