# Peer review of "Plant-Based Diets for Cardiovascular Safety and Performance in Endurance Sports"

_nutrients, 2019, doi:10.3390/nu11010130_

Reviewer 1 Report

The authors of the manuscript reviewed some evidence on plant-based diets and cardiovascular/physiological risk factors, and discussed the possible relevance to performance for endurance athletes. The topic is interesting, but the implications of the research are unclear. Please see below for some major and minor comments.

 Major comments:

1)   The aim of this manuscript has not been clearly stated, but seemed to be to provide a review of how plant-based diets might affect cardiovascular risk factors and subsequently athletic performance in endurance sports. However, the importance of studying plant-based diets specifically, rather than other healthy or balanced dietary patterns, in performance has not been explained.

 2)   The authors did not adopt a systematic approach of reviewing the available evidence, and therefore it was not possible to reach a balanced conclusion on the topic. Large-scale observational evidence mostly reported that regular physical activity was associated with lower risk of cardiovascular diseases or cardiovascular risk factors (PMID: 28097313, 29345215). Without a clear definition of the population of interest (and the definition of endurance sports), the public health implications of the work could be misleading.

 3)   In discussing the contribution of plant-based diets to the various physiological factors, the authors mostly quoted results from short-term trials. Although trial evidence could be considered as superior to observational evidence, many dietary trials were of very short time frames and consisted of small participant numbers. Therefore, it would be preferable if evidence from large-scale observational cohorts, especially studies with large proportion of long-term vegans could be considered as well.

 4)   The authors stated that the terms plant-based diets and vegans are used interchangeably in the manuscript. However, many of the referenced studies included or were exclusively of vegetarians, who consumed dairy. Therefore, the definition of plant-based diets should be clarified.

 Minor comments:

 5)   It would be helpful to the reader if  the headings of 3. Leaner body mass, 4. Facilitating glycogen storage and so on, could be relabelled as 2.1 and 2.2, to clearly indicate that these are examples of the role of diet in athletic performance.

Author Response

Thank you for these helpful comments. We have addressed all of them below and made the appropriate adjustments in the manuscript.

Responses to Reviewer 1:

Based on this comment, we added in a "Purpose" statement to the abstract and put in more context in the introduction of the paper as well.

We appreciate this observation and suggestion.The examples provided by the reviewer are good studies but not exactly to our points. These studies show how exercise helps with heart disease. We are attempting to explain how plant-based diets lead to a healthier heart which in turn improves one's physical performance. As opposed to a systematic review, we are examining studies that may help answer a hypothesis ... Vegan athletes may consume diets that support overall better heart health.

We agree. We have added the findings of larger observational studies to our paper and the reference list.

Thank you for this observation. We have added clarification to the initial explanation of terminology and clarified when necessary throughout the paper.

We appreciate this suggestion, and we have implemented it. 

Reviewer 2 Report

This is a very interesting paper, well documented and clear.

I have only one major point to do:

- the authors only talk about macroutrients, but do not consider their quality. I think that two points would need to be further stressed:

- it is difficult to disentangle the effect of eating meat (particularly if red) from the effect of eating additives... Processed meat is very common in several settings and the effect of added "poisons" should in my opinion be mentioned.

- the problem of pesticides, and of preserving agents on the contrary, which may be relevant in plant based diets, should also be mentioned, as it could offset the advantage of an healthier diet. 

Author Response

Thank you for these helpful comments. We have addressed all of them below and made the appropriate adjustments in the manuscript.

Responses to Reviewer 2: 

We appreciate these suggestions, and we added more information about meat and different types of meat to our inflammatory section of the paper. 

The reviewer raises an interesting and important point about pesticide residues in plant food, which has not been addressed to a substantial degree in the literature, and we did not have a useful comment on this to add to  the manuscript. We note that it is a potential issue for both vegetarians and nonvegetarians, as pesticide residues may also occur in animal-derived products. On a related note, a 1981 study found reduced chemical pollutants in vegetarians, using breast milk samples as an indicator (Hergenrather J, et al. Pollutants in breast milk of vegetarians. Lancet. 1981;304:792.). But we are not aware of other literature on this point.  

Round  2

Reviewer 1 Report

Thank you for considering and addressing my previous comments. I have the following minor suggestion:

Line 43 and reference 6 (Kwok et al 2014. Int J Cardiol): the RR for cerebrovascular disease in this meta-analysis was not significant with wide confidence intervals (RR 0.71, 0.41-1.20 in Adventist studies and RR 1.05, 0.89-1.24 in non-adventist studies). Therefore, I would suggest removing the mention of 29% reduced risk for cerebral vascular disease events from the text, since the evidence is not convincing and the cited estimate applies only to Adventist populations. Similarly, in the same sentence it should be clarified that 40% lower risk of CHD referred specially to Adventist studies, since the pooled result from non-adventist studies (which might be more representative of the general population in comparison) was more modest.

Author Response

We very much appreciate your time spent considering our revision. Thank you for this recent suggestion. We have removed the Kwok reference from the manuscript. 

Reviewer 2 Report

This is a nice review, even if non systematic.

the authors have correctly integrated the comments and remarks.

Author Response

We very much appreciate your time on our submission and for making our paper stronger.